# Formation of Polyaniline and Polypyrrole Nanocomposites with Embedded Glucose Oxidase and Gold Nanoparticles

**DOI:** 10.3390/polym11020377

**Published:** 2019-02-20

**Authors:** Natalija German, Almira Ramanaviciene, Arunas Ramanavicius

**Affiliations:** 1Department of Immunology, State Research Institute Center for Innovative Medicine, Santariskiu 5, LT-08406 Vilnius, Lithuania; natalija.german@imcentras.lt; 2Department of Physical Chemistry, Faculty of Chemistry and Geosciences, Vilnius University, Naugarduko 24, LT-03225 Vilnius, Lithuania; 3NanoTechnas–Centre of Nanotechnology and Materials Science, Faculty of Chemistry and Geosciences, Vilnius University, Naugarduko 24, LT-03225 Vilnius, Lithuania; almira.ramanaviciene@chf.vu.lt; 4Division of Materials Science and Electronics, State Scientific Research Institute Center for Physical Sciences and Technology, Savanorių ave. 231, LT-02300 Vilnius, Lithuania

**Keywords:** Glucose oxidase, Gold nanoparticles, Polyaniline, Polymeric nanocomposites, Polymerization, Polypyrrole

## Abstract

Several types of polyaniline (PANI) and polypyrrole (Ppy) nanocomposites with embedded glucose oxidase (GOx) and gold nanoparticles (AuNPs) were formed by enzymatic polymerization of corresponding monomers (aniline and pyrrole) in the presence of 6 and 13 nm diameter colloidal gold nanoparticles (AuNPs_(6nm)_ or AuNPs_(13nm)_, respectively) or chloroaurate ions (AuCl_4_^−^). Glucose oxidase in the presence of glucose generated H_2_O_2_, which acted as initiator of polymerization reaction. The influence of polymerization bulk composition and pH on the formation of PANI- and Ppy-based nanocomposites was investigated spectrophotometrically. The highest formation rate of PANI- and Ppy-based nanocomposites with embedded glucose oxidase and gold nanoparticles (PANI/AuNPs-GOx and Ppy/AuNPs-GOx, respectively) was observed in the solution of sodium acetate buffer, pH 6.0. It was determined that the presence of AuNPs or AuCl_4_^−^ ions facilitate enzymatic polymerization of aniline and pyrrole.

## 1. Introduction

Intensive studies and the application of conducting polymers (CPs), such as polyaniline (PANI) and polypyrrole (Ppy), have been developed extensively over the past few decades [1,2,3,4,5]. Both PANI and Ppy have conjugated π-orbitals in their polymeric backbone. CPs are mostly formed as thin films [6,7], colloids [8,9,10,11], nanocomposites [12], nanowires [12] or water-soluble materials [6,13]. CPs, being electrically conductive and mechanically soft, can interface effectively with some electroactive tissues and cells [3]. Moreover, CPs are the very promising materials for biomedical and bioanalytical applications (e.g., for biosensor design), for rechargeable batteries, light-emitting diodes, electrochromic display devices, solar cells, in photothermal therapy, in corrosion prevention [2,3,12,14,15,16], because CPs act as transducers [3,17]. In such applications, conducting polymer layers of different morphology and thickness are applied [1,5,18].

Usually, CPs are synthesized by chemical [11,13,19,20,21,22,23,24] or electrochemical [2,6,7,15,16,20,22,25,26] polymerization of a monomer characterized by pronounced electron donor properties [6,11,27,28,29]. Previously it was reported that horseradish peroxidase can be used for the formation of some CPs [1,8,9,10,14,28], later our team has introduced glucose oxidase (GOx) based formation of Ppy [8,11], PANI [9] and some other CPs. Advantages of enzymatic polymerization are based on possible control of the reaction kinetics and the simplicity of this mostly one-step-based process [9,10,30]. During GOx-based enzymatic formation of PANI and Ppy polymerization reaction is initiated by H_2_O_2_, which is produced during glucose oxidase catalysed oxidation of glucose [8,9,28,31]. Various forms of PANI including leucoemeraldine, emeraldine (E-PANI) and pernigraniline (P-PANI) can be formed during oxidation/reduction of PANI-based structures [19,20,30,31].

Hybrid materials based on CPs containing inorganic nanoparticles (e.g., gold or silver nanoparticles) have unique properties and, therefore, they are promising for the application in electronics, optoelectronics and in bio-devices [12,13,27,32,33,34,35,36,37]. It was determined that the coordination of heteroatoms are responsible for the heterocycle linkage, because the presence of oxygen or nitrogen increases the adsorption-ability of polymers toward gold nanoparticles (AuNPs) [32]. First Au(III) is able to be reduced into Au(I) form, which can form a nuclei for nanoparticle and only then Au(I) is reduced to the state of zero-valent gold (Au(0)) [36] Au(I) is able to activate inert alkenes and 1,3-dienes and the formation of an Au(I)-diene complex is possible [32]. Similar interaction contributes to the formation of bonding between the gold nanoparticles and the conjugated diene-based moieties of CPs chains [32]. The second reduction step from Au(I) to Au(0) is responsible for the formation of clusters [36]. Therefore, both conducting polymer and AuNPs play a role in the formation of such nanocomposites [36,37]. Oxidized portions of polymer chain are related to an interaction with gold nanoparticles and its oxidative properties. AuNPs are able speed-up the polymerization of monomers [36,37]. Usually polymers/AuNPs-based nanocomposites are formed in the form of insoluble powder.

In this research enzymatic polymerization based formation of polyaniline and polypyrrole nanocomposites with embedded glucose oxidase and gold nanoparticles is described. Polyaniline and polypyrrole based composite formation rate in the presence of different AuNPs, at different pHs of solution was evaluated by UV/Vis spectroscopy. 

## 2. Materials and Methods

### 2.1. Materials

Glucose oxidase (EC 1.1.3.4, type VII, from *Aspergillus niger*, 201 units mg^−1^ protein) and D-(+)-glucose were purchased from Fluka (Buchs, Switzerland) and Carl Roth GmbH+Co.KG (Karlsruhe, Germany), respectively. Before the investigations glucose solution was allowed to stay overnight for the formation of equilibrium between α and β optical isomers. Tetrachloroauric acid (HAuCl_4_·3 H_2_O) was obtained from Alfa Aesar GmbH&Co KG (Karlsruhe, Germany), tannic acid – from Carl Roth GmbH + Co (Karlsruhe, Germany) and sodium citrate – from Penta (Praha, Czech Republic). All solutions were prepared using deionized water purified with water purification system Millipore S.A. (Molsheim, France). The solution of sodium acetate (SA) buffer (0.05 mol L^−1^ CH_3_COONa·3H_2_O) with 0.1 mol L^−1^ KCl was prepared by mixing of sodium acetate trihydrate and potassium chloride, which were obtained from Reanal (Budapest, Hungary) and Lachema (Neratovice, Czech Republic). Aniline and sodium hydroxide (NaOH) were purchased from Merck KGaA (Darmstadt, Germany), pyrrole – from Acros Organics (New Jersey, NJ, USA) and hydrochloric acid (HCl) – from Sigma-Aldrich (Saint Louis, MO, USA). All chemicals used in experiments were either analytically pure or of highest quality. Polymers were filtered before each measurement through 5 cm column filled by Al_2_O_3_ powder to remove coloured components. All solutions were stored between measurements at +4 °C. The synthesis of AuNPs_(6nm)_ and AuNPs_(13nm)_ was performed according to methodology reported previously [38].

### 2.2. Formation and Separation of PANI- and Ppy-Based Nanocomposites with Embedded Glucose Oxidase and Gold Nanoparticles (PANI/AuNPs-GOx or Ppy/AuNPs-GOx)

PANI/AuNPs-GOx and Ppy/AuNPs-GOx composites were formed at room temperature (+20 ± 2 °C) in darkness in solution containing 0.05 mol L^−1^ of glucose, 0.50 mol L^−1^ of aniline or pyrrole, 0.75 mg mL^−1^ of GOx and 26.0, 3.6 nmol L^−1^ of AuNPs_(6nm)_ or AuNPs_(13nm)_ or 0.6 mmol L^−1^ of HAuCl_4_. Formed nanocomposites were separated from the synthesis solution by a centrifugation (8 min, 16.1 × 10^3^ g), then they were 2 times washed with 0.05 mol L^−1^ sodium acetate, pH 6.0 and were collected by centrifugation. Separated and washed PANI/AuNPs-GOx and Ppy/AuNPs-GOx nanocomposites were re-suspended in SA buffer, pH 6.0 and used for further investigations, which were performed at room temperature. 

### 2.3. The Optimization of PANI/AuNPs-GOx and Ppy/AuNPs-GOx Synthesis

The presence of gold in the polymerization solution, the pH of medium and the time of enzymatic polymerization have an influence on the synthesis of polymer’s nanoparticles. The investigations of the influence of AuNPs or chloroaurate ions (AuCl_4_^−^) were evaluated after enzymatic formation of nanocomposites in 0.05 mol L^−1^ SA buffer, pH 6.0, 0.05 mol L^−1^ of glucose, 0.50 mol L^−1^ of aniline or pyrrole, 0.75 mg mL^−1^ of GOx and 26.0, 3.6 nmol L^−1^ of AuNPs_(6nm)_ or AuNPs_(13nm)_ or 0.6 mmol L^−1^ of HAuCl_4_.

The polymerization in bulk solution consisting 0.05 mol L^−1^ of glucose, 0.75 mg mL^−1^ of GOx, 0.50 mol L^−1^ of aniline or pyrrole monomers and AuNPs_(6nm)_, AuNPs_(13nm)_ or AuCl_4_^−^ was studied at different pHs ranging from 1.0 to 12 after 2 days. 0.1, 0.01 or 0.001 mol L^−1^ of HCl, 0.05 mol L^−1^ of SA buffer, pH 6.0, and 1 × 10^−6^, 1 × 10^−4^ or 0.01 mol L^−1^ of NaOH solutions were used for the preparation polymerization bulk solutions in pH range between 1.0 and 12.0. The duration of polymerization reaction was from 5 hours until 12 days in 0.05 mol L^–1^ SA buffer, pH 6.0, 0.05 mol L^−1^ of glucose, 0.50 mol L^−1^ of aniline or pyrrole, 0.75 mg mL^–1^ of GOx and 26.0 nmol L^–1^ of AuNPs_(6nm)_ or 0.6 mmol L^−1^ HAuCl_4_. 

### 2.4. The Monitoring of PANI/AuNPs-GOx and Ppy/AuNPs-GOx Synthesis by UV/Vis Spectroscopy

The polymerization course was evaluated by UV/Vis spectroscopy. Optical absorbance of PANI/AuNPs-GOx and Ppy/AuNPs-GOx solutions was monitored by UV/Vis spectrometer Lambda 25 (Shelton, CT, USA) at the wave length range of 233–1000 nm. The investigations were performed at room temperature in plastic disposable cuvettes of 1 cm optical path length by the registration of UV/Vis spectra. After 5–10 minutes after the engagement of polymerization reaction at +20 ± 2 °C optical absorbance of PANI/AuNPs-GOx and Ppy/AuNPs-GOx solutions was registered. SigmaPlot software 12.5 was used to evaluate the data of UV/Vis measurements. The main steps of PANI/AuNPs-GOx and Ppy/AuNPs-GOx nanocomposite formation during enzymatic polymerization are presented in Figure 1.

## 3. Results and Discussion

CPs such as polyaniline and polypyrrole are often used in catalysis [10,20] or in bio-electrocatalysis for the immobilization of various biological molecules [1]. There are some expectations that the incorporation of elementary gold into polymers can improve charge transfer from enzymes to the electrode. It is predicted that novel nanocomposites based on AuNPs and CPs could provide various interesting characteristics suitable for their application in biosensorics. Therefore, the main aim of present study was the formation of PANI/AuNPs-GOx and Ppy/AuNPs-GOx nanocomposites with incorporated enzyme–GOx. As it is seen from Figure 1 in the presence of glucose and dissolved oxygen enables GOx to generate lactone of gluconic acid and hydrogen peroxide, which is necessary to initiate pyrrole or aniline polymerization reaction [28]. 

After the mixing of all reagents used for PANI and Ppy formation in the presence of AuNPs_(6nm)_ and AuNPs_(13nm)_ in SA, pH 6.0, any optical absorbance peaks were registered in vis-spectra. Although, in the presence of AuCl_4_^−^ instead of AuNPs the solution of pyrrole almost instantly becomes intensively black, what is clear evidence of polymerization reaction. As it is seen from Figure 2 after 3 days lasting formation of PANI/AuNPs-GOx and Ppy/AuNPs-GOx nanocomposites in the presence of AuNPs_(6nm)_, AuNPs_(13nm)_ or HAuCl_4_ become brown (in the presence of PANI) and black (in the presence of Ppy) sediments. Such colours are the most characteristic for the delocalised *π*-electron systems of CPs [28]. 

The intensity of colour of synthesized polymers also depends on the pH of initial polymerization bulk solution. The range of tested pHs was between 1.0 and 12. As it is seen from Figure 3A,B,C and 3D the UV/Vis spectra of PANI/AuNPs-GOx nanocomposites after 2-days lasting polymerization exhibit two basic wide peaks at *λ* = 285 nm and *λ* = 570 nm if AuNPs_(6nm)_ are added into polymerization bulk solution and only one wide peak at *λ* = 550 nm appears if AuNPs_(13nm)_ are added into polymerization bulk solution. If polymerization is performed in the presence of AuCl_4_^−^ the peak maximum appears at *λ* = 450 nm. This fact is an agreement with other researches [13,24,37]. As it was described by other researches the optical absorbance peak at *λ* = 330 nm appears due to *π*-*π** transition of the benzenoid ring and another peak at *λ* = 440 nm—due to polaron *π*-transition [13,24]. Results clearly demonstrate that the shift of optical absorption peak maximum depends on the composition of polymerization bulk solution and the duration of polymerization. Optical absorbance at *λ* = 330 nm is indication of 5,10- dihydrophenazin formation, which is cyclic aniline dimer [21]. *π*-*π** transition is associated with the delocalization of electrons from benzene rings on nitrogen atoms of aniline [6,21,29]. The further oxidation of 5,10-dihydrophenazine is leading to the formation of phenazylium radical cation with the absorbance maximum at *λ* = 436 nm. It is due to electron‘s excitation of benzenoid highest occupied molecular orbital to quinoid lowest unoccupied molecular orbital and indicates the formation of emeraldine forms of PANI [6,13,21,24,29]. The absorbance at 570 nm is indicating the oxidation of E-PANI into P-PANI, according to other researches the optical absorbance peak of PANI-based sample, which consist of up to 50% of E-PANI and/or up to 100% of P-PANI, is in the range from 400 until 800 nm [39].

As it is seen from spectra presented in Figure 3A,B,C and D at pH values, which are lower than 6.0, PANI/AuNPs-GOx has a weak optical absorbance wave in the range from λ = 442 nm to *λ* = 570 nm. By the increase of pH value of polymerization solution, the polaron-related peak at λ = 420 nm gradually disappears and a strong absorbance due to excision transition of the quinoid rings at *λ* = 560–600 nm is emerging. At the same time in strong alkaline polymerization solution optical absorbance at *λ* = 257 nm and *λ* = 320 nm, which is observed due to ð-ð* transitions of the benzenoid rings, has increased [10]. The absorbance wave at *λ* = 442–570 nm and the peak at *λ* = 450 nm in the presence of AuNPs and AuCl_4_^–^, respectively, were evaluated to assess the formation of PANI/AuNPs-GOx nanocomposites during the polymerization. Synthesized PANI/AuNPs-GOx nanocomposites were formed in the E-PANI, which has been confirmed by the appearance of brown colour.

The influence of pH value of polymerization solution on synthesized PANI/AuNPs-GOx nanocomposites is presented in Figure 3E. It was determined that very low pH value (pH < 1.0) is not suitable for the formation of E-PANI, because H^+^ ions are produced during polymerization reaction. As seen from the presented curves (Figure 3E), the increase in the pH of the polymerization solution was changed from 2.0 to 6.0 optical absorbance to 0.281 to 0.274 a.u., if AuNPs_(6nm)_ were present in the polymerization solution; and from 0.256 to 0.223 a.u. in the case where AuNPs_(13nm)_ were present in the polymerization solution and from 0.813 to 0.283 a.u. in the case where AuCl_4_^−^ was present in the polymerization solution. Synthesis of some branched polyaniline chains is expected due to polaron transitions, which are reflected in spectra [10]. When the pH of polymerization solution has been increased until pH 12, then the decrease of optical absorbance until 0.096 and 0.605 a.u. was observed for polymerization solution containing AuNPs_(6nm)_ or AuCl_4_^−^, respectively. When the pH of polymerization solution containing AuNPs_(13nm)_ was equal to pH 10, then optical absorbance decreased until 0.217.

In research performed by another authors it was predicated that aniline in polymerization bulk solution remain mobile and the synthesis of polymers could be proceeded without any enzyme [10]. Moreover, during recent our investigations it was observed that auto-polymerization is observed only for polyaniline in the presence of AuCl_4_^−^ in pH range from 2.0 to 12. Meantime, the auto-polymerization of PANI in the presence of AuNPs_(6nm)_ and AuNPs_(13nm)_ was observed only in pH ranges from 1.0 until 3.0. The auto-polymerization of aniline in polymerization solution of pH 2.0 with AuNPs_(6nm)_, AuNPs_(13nm)_ or AuCl_4_^−^ was 1.83, 1.79 and 2.78 times slower than that in polymerization solution with the same gold-based compounds and GOx. For enzymatic formation of PANI/AuNPs-GOx nanocomposites neutral polymerization solution is required due to best enzymatic activity and stability of GOx [9].

The influence of pH value of polymerization solution on optical absorbance during the formation of Ppy/AuNPs-GOx nanocomposites was investigated in the next stage of our investigations. During 2-days lasting enzymatic formation of Ppy/AuNPs-GOx nanocomposites at pH values ranging from 1.0 until 12 in the presence of AuNPs or AuCl_4_^−^ ions were intensively black coloured and solutions above a sediment consisting of aggregated Ppy/AuNPs-GOx become dark green. Dark colour of formed Ppy-based nanocomposites is characterized for advanced *π*-conjugated system [28]. As it is seen from Figure 4A–C the absorbance was observed for spectra of formed Ppy/AuNPs-GOx nanocomposites at *λ* = 265 nm, *λ* = 312–361 nm, *λ* = 414 nm and *λ* = 560 nm, what is in an agreement with spectra of chemically synthesized polypyrrole [23,33,34]. The peak at *λ* = 265 nm is associated with the *π*-conjugated structure of pyrrole rings, at *λ* = 312–361 nm and at *λ* = 414 nm—with *π*-*π** transition of the polypyrrole‘s chain [23,33,34]. The peak at *λ* = 560 nm is attributed to a transition from the valence bond to the antibonding polaron- or bipolaron-state and reveals that ultra-small clusters of AuNPs and Ppy are formed [23,33,34]. By the increase of pH value the optical absorbance at *λ* = 265 nm and *λ* = 312–361 nm was changed just slightly but at *λ* = 414 nm and *λ*
*=* 560 nm changes of optical absorbance were very significant. The absorbance at *λ* = 414 nm is characteristic for the early stage of pyrrole oxidation.

As it is seen from Figure 4A–C the most rapid increase of optical absorbance was observed in strongly acidic solution at pH 1.0. Therefore this pH can be considered as the most optimal for the formation of Ppy/AuNPs-GOx. But like in the case of polyaniline formation the pH values over 1.0 are not suitable for enzymatic polymerization, because the activity and the stability of GOx at such extreme pHs are not sufficient enough. As it is seen from curves presented in Figure 4D, by the increase of pH value of polymerization solution from 2.0 until 6.0, optical absorbance at *λ* = 480 nm has been changed from 0.230 to 0.403 a.u. in the presence of AuNPs_(6nm)_, from 1.33 × 10^−3^ to 0.141 a.u. – of AuNPs_(13nm)_ and from 0.248 until 0.608 a.u.– of AuCl_4_^−^. In polymerization solution of pH 12, the optical absorbance at *λ* = 480 nm decreased down until 0.169, 3.04 × 10^−3^ and 0.144 a.u. in the case of AuNPs_(6nm)_, AuNPs_(13nm)_ and AuCl_4_^−^, respectively. Such decrease of optical absorbance by increasing pH until 12 is an agreement with investigations of Ppy chemical synthesis in polymerization bulk solution without any gold-based compounds [11].

When polymerization compounds are mixed, then the polymerization of pyrrole and the reduction of AuCl_4_^−^ into Au^0^ occurs very quickly and simultaneously darkening of the solution is observed [33]. Similar effect was observed in the polymerization bulk solution containing AuNPs. It should be taken into account that the formation of Ppy is possible in the solution without any chemical oxidant or enzyme [11]. It was determined that auto-polymerization in the presence of AuNPs or AuCl_4_^−^ ions and in the absence of GOx was observed at pHs ranging from 2.0 to 3.0 during the formation of Ppy/AuNPs_(6nm)_ and Ppy/AuNPs_(13nm)_ nanocomposites and from 1.0 to 3.0 during the formation of Ppy/AuNPs_(AuCl4_^−^_)_ nanocomposites. The auto-polymerization of pyrrole in the presence of AuNPs_(6nm)_, AuNPs_(13nm)_ and AuCl_4_^−^ in polymerization bulk solution at pH 2.0 was 5.47, 1.31 and 2.89 times slower, if compared with the enzymatic formation of Ppy/AuNPs-GOx nanocomposites.

Significant differences of the optical absorbance were detected in the absence and in the presence of GOx, what revealed a high impact of the enzyme to the polymerization rate of monomers [8]. Changes of absorbance during the auto-polymerization were not observed in polymerization bulk solution at pH values in the interval of pH 5.5 and 6.0 and in alkaline medium at pH 8.0, 10 and 12. Hence, it was concluded that the auto-polymerization rate in the absence of GOx is not high when the pH of polymerization solution is higher than 3.0, what is an agreement with another investigations [8].

Enzymatic formation of PANI/AuNPs-GOx and Ppy/AuNPs-GOx depends on source of gold, which was AuNPs_(6nm)_, AuNPs_(13nm)_ or AuCl_4_^−^ ions. AuNPs are forming seeds for polymerization and AuCl_4_^−^ ions are acting as oxidators, which initiates the polymerization of CPs [40] and forms Au^0^ forms AuNPs, which act as seeds for polymerization. In the presence of HAuCl_4_ the reduction of AuCl_4_^−^ and the formation of oligoaniline is the dominant reaction in the polymerization bulk solution [13]. The reaction between AuCl_4_^−^ ions and PANI is following by next steps. First of all, polyaniline coordinates Au(III) via an amine N, which is substituting Cl^−^ in AuCl_4_^−^, because four Cl^−^ anions are released into polymerization solution during PANI formation [13,24]. Then, Au(III) ions by two electrons are oxidizing PANI benzenoid group to quinoid group. Formed AuCl_2_^−^ ion is able to coordinate with an imine N of the quinoid group through electron donation from N to Au and formation of N–H----Cl^−^ type hydrogen bonds, therefore AuCl_4_^−^ ion is taking active and important role in the formation of PANI/AuNPs based nanocomposites [24].

The influence of gold compounds and polymerization rate of polyaniline and polypyrrole nanocomposites formation was investigated in 0.05 mol L^−1^ sodium acetate buffer, pH 6.0, which was chosen as the most optimal for mentioned gold/polymers enzymatic polymerization. Higher value of optical absorbance of polyaniline was characterized for polymerization solution with AuCl_4_^−^ and achieved 0.674 a.u. (Figure 5). However, in the presence of AuNPs_(6nm)_ and AuNPs_(13nm)_ the polymerization rate of polyaniline formation was very similar and optical absorbance of 0.274 and 0.256 a.u was registered, respectively. In the presence of AuCl_4_^−^ the formation of PANI/AuNPs_(AuCl4_^−^_)_-GOx nanocomposites was 2.46 and 2.63 times faster than that in polymerization bulk solution containing AuNPs_(6nm)_ and AuNPs_(13nm)_. The same influence of gold-based compounds was observed during the polymerization of pyrrole. As it is presented in Figure 5 optical absorbance of Ppy/AuNPs_(AuCl4_^−^_)_-GOx nanocomposites (0.608 a.u.) was 1.51 and 4.31 times higher if compared with results obtained in polymerization bulk solution containing AuNPs_(6nm)_ (0.403 a.u.) and AuNPs_(13nm)_ (0.141 a.u.). It is seen that the formation rate of PANI/AuNPs_(AuCl4_^−^_)_-GOx and PANI/AuNPs_(13nm)_-GOx nanocomposites was 1.11 and 1.82 times higher if compared with of Ppy/AuNPs_(AuCl4_^−^_)_-GOx and Ppy/AuNPs_(13nm)_-GOx (Figure 5). Meantime, optical absorbance of PANI/AuNPs_(6nm)_-GOx formed nanocomposites colloidal solution was 1.47 times lower than that of Ppy/AuNPs_(6nm)_-GOx nanocomposites colloidal solution. It could be explained by short duration of polymerization.

During control experiments we tested: (i) the polymerization bulk solutions of PANI and Ppy, which did not contain AuNPs and (ii) the solution, which contained glucose, GOx, and AuNPs but did not contain any monomers (aniline or pyrrole), which are required for the formation of polymers. As it is presented in Figure 5 the optical absorbance of PANI/AuNPs_(AuCl_4__^−^_)_-GOx and Ppy/AuNPs_(AuCl_4__^−^_)_-GOx nanocomposites was 2.14 and 3.52 times higher if compared with that registered for PANI/GOx (0.315 a.u) [9] and Ppy/GOx (0.173 a.u.). The formation rate for PANI/AuNPs_(6nm)_-GOx and PANI/GOx was very similar; for Ppy/AuNPs_(6nm)_-GOx was 2.33 times higher than that obtained for Ppy/GOx nanocomposites. The formation rate of PANI/AuNPs-GOx and Ppy/AuNPs-GOx nanocomposites in comparison to that of PANI/GOx and Ppy/GOx is significantly higher, because AuNPs speed-up the agglomeration of formed oligomers and the most probably facilitate polymerization reaction [36,37]. As it was expected any observable composites were formed in the solution containing glucose, GOx and AuNPs in the absence of PANI and Ppy. It means that in enzymatic polymerization at least three compounds (glucose, GOx and aniline or pyrrole) are required for the formation of composite materials. Therefore, for further investigations of PANI/AuNPs-GOx and Ppy/AuNPs-GOx nanocomposites formation AuNPs_(6nm)_ and AuCl_4_^−^ were chosen, which were both considered to be the most suitable for the formation of these aniline- and pyrrole-based nanocomposites [30].

One of the most important parameters of polymerization process is the duration of the polymerization. Thus, the variation of duration of enzymatic polymerization enabled to achieve the optimal polymerization performance. The spectrometric investigation of enzymatic polymerization duration on formation of PANI/AuNPs-GOx and Ppy/AuNPs-GOx nanocomposites was performed in SA buffer, pH 6.0, in the presence of aniline or pyrrole and of AuNPs_(6nm)_ or 0.6 mmol L^−1^ AuCl_4_^−^, correspondingly. Visible aggregation of polyaniline and polypyrrole in the presence of AuNPs and AuCl_4_^−^ were observed after 2 and 4.5 days of enzymatic polymerization. 

As it is presented in Figure 6A,B one well measurable peak at *λ* = 450 nm was achieved after 2 and 1 days for PANI/AuNPs_(6nm)_-GOx and PANI/AuNPs_(AuCl_4__^−^_)_-GOx nanocomposites formation. This optical absorbance band appears due to polaron transition from benzenoid group at highest occupied molecular orbital to quinoid group at lowest unoccupied molecular orbital, which indicates presence of emeraldine form of PANI/AuNPs-GOx nanocomposites and is in agreement with results by another authors [13,21,24,27,30].

As it is seen from Figure 6A,B the reduction of colloidal gold and polyaniline was observed until 10^th^ day of polymerization. It is indicated, that the highest value of optical absorbance for PANI/AuNPs_(6nm)_-GOx (1.05 a.u.) and PANI/AuNPs_(AuCl_4__^−^_)_-GOx (1.24 a.u.) was achieved after 10 days lasting enzymatic polymerization. The value of optical absorbance was 11.1 and 3.17 times higher than that after 5 hours lasing polymerization, respectively for PANI/AuNPs_(6nm)_-GOx and PANI/AuNPs_(AuCl_4__^−^_)_-GOx. Moreover, long polymerization’s duration enables to synthesize larger PANI/AuNPs-GOx nanocomposites. Therefore, for the formation of smaller nanocomposites short duration of enzymatic polymerization is recommended. As it is seen from Figure 6C, by the increase of polymerization duration from 5 hours to 5 days in the presence of AuNPs_(6nm)_ and 0.6 mmol L^−1^ AuCl_4_^−^ optical absorbance at *λ* = 450 nm increased from 0.095 to 0.517 a.u. and from 0.391 to 1.11 a.u, respectively. The increase of optical absorbance by 5.44 and 2.84 times after 5 days for PANI/AuNPs_(6nm)_-GOx and PANI/AuNPs_(AuCl_4__^−^_)_-GOx formed nanocomposite colloidal solutions was observed. The appearance of this peak in the presence of small nanocomposites in polymerization solution could be explained by fast movement of free electrons to the surface of AuNPs-based nanocomposites, during which electrons are scattered and quickly lose the coherence. In other words, the surface plasmon resonance bandwidth increases by the decrease of AuNPs size [37].

Enzymatic polymerization of pyrrole started very fast and after 5 hours of initiation of polymerization by addition of AuNPs_(6nm)_ or 0.6 mmol L^−1^ AuCl_4_^−^ to the reactants UV/Vis spectra (Figure 6D,E) showed two wide waves of an absorbance between *λ* = 448 nm and *λ* = 550 nm and between *λ* = 460 nm and *λ* = 590 nm, respectively. It was investigated that firstly the absorbance maximum of Ppy/AuNPs_(6nm)_-GOx and Ppy/AuNPs_(AuCl_4__^−^_)_-GOx nanocomposites were achieved at *λ* = 550 nm and *λ* = 590 nm, respectively. After one day of enzymatic polymerization, these waves almost disappeared and new bands of π-π* transition of Ppy/AuNPs chain and a polaron transition similar to that of an ordinary Ppy/AuNPs_(6nm)_-GOx and Ppy/AuNPs_(AuCl_4__^−^_)_-GOx at *λ* = 448 nm and *λ* = 460 nm were observed more clearly [11,28,33,34]. The absorbance at this wave length is related to electronic transitions associated with polarons and/or bipolarons [11]. It was concluded that core/shell structure of Ppy/AuNPs-GOx nanocomposites was formed. The similar observation was reported by another authors during chemical and electrochemical polymerization of pyrrole [28,33,34]. Figure 6F illustrates that in the presence of AuNPs_(6nm)_ or 0.6 mmol L^−1^ AuCl_4_^−^ in polymerization bulk solution, a dark pyrrole was relatively rapidly synthesized and optical absorbance of formed nanocomposite colloidal solution, which was measured at two polymerization periods (after 5 h and 10 days) increased from 0.137 to 0.781 a.u. and from 0.159 to 1.05 a.u., respectively. It is seen that optical absorbance, which is characteristic for Ppy/AuNPs_(6nm)_-GOx and Ppy/AuNPs_(AuCl_4__-_)_-GOx nanocomposites, after 10 days of polymerization was 5.70 and 6.60 times higher than that registered after 5 hours. This indicates the formation of polypyrrole oligomers [11]. From 5 hours until 4.5 days increased the duration of enzymatic formation of Ppy/AuNPs_(6nm)_-GOx and Ppy/AuNPs_(AuCl_4__^−^_)_-GOx nanocomposites has changed optical absorbance of polymerization bulk solution until 0.494 and 0.538 a.u, respectively. It was determined that after 4.5 days of enzymatic polymerization the value of optical absorbance for Ppy/AuNPs_(6nm)_-GOx and Ppy/AuNPs_(AuCl_4__-_)_-GOx nanocomposites increased by 3.61 and 3.38 times, respectively. The decrease of absorbance value after 12 days of formation of nanocomposites is explained by aggregation of nanocomposites due their limited solubility.

It was demonstrated that synthesized PANI/AuNPs-GOx and Ppy/AuNPs-GOx nanocomposites were similar to that obtained by chemical or electrochemical methods [13,33,34] and may be successful formatted enzymatically. Optical absorbance of PANI/AuNPs_(6nm)_-GOx and PANI/AuNPs_(AuCl_4__^−^_)_-GOx nanocomposites was 1.05 and 2.06 times higher compared with that of Ppy/AuNPs_(6nm)_-GOx and Ppy/AuNPs_(AuCl_4__^−^_)_-GOx.

The polymerization rate (*V*) was evaluated using formula *V* = (tgα), where tgα was calculated from Figure 6C,F by using an estimated/approximated absorbance (1.56 and 2.98 a.u. for PANI/AuNPs_(6nm)_-GOx and PANI/AuNPs_(AuCl_4__^−^_)_-GOx; 1.73 and 2.23 a.u. for Ppy/AuNPs_(6nm)_-GOx and Ppy/AuNPs_(AuCl_4__^−^_)_-GOx) formed when 100% of the monomers present in the solution are converted into polymers. Initial polymerization duration in polymerization bulk solution containing aniline with AuNPs_(6nm)_ or AuCl_4_^−^ was determined as 0.0026 and 0.0022 mol L^−1^ day^−1^, respectively; for solution containing pyrrole with AuNPs_(6nm)_ or AuCl_4_^−^ – 0.0031 and 0.0029 mol L^−1^ day^−1^, respectively. Initial formation rate of PANI/AuNPs_(6nm)_-GOx and PANI/AuNPs_(AuCl_4__^−^_)_-GOx nanocomposites was 1.19 and 1.32 times slower than that of Ppy/AuNPs_(6nm)_-GOx and Ppy/AuNPs_(AuCl_4__^−^_)_-GOx, respectively. However, AuNPs do not form the uniform and well interconnected structure because nanocomposites are embedded between polymer backbone [37].

## 4. Conclusions

Enzymatic polymerization of aniline and pyrrole monomers in the presence of AuNPs and AuCl_4_^−^ was used to synthesize PANI/AuNPs-GOx and Ppy/AuNPs-GOx nanocomposites. The formation of emeraldine base’s form of PANI and conducting Ppy nanocomposites with embedded glucose oxidase and AuNPs was determined by the evaluation of visible spectra, which showed characteristic waves of optical absorbance at *λ* = 450 nm and *λ* = 480 nm, respectively. It was determined that 26.0 nmol L^−1^ of AuNPs_(6nm)_ and 0.6 mmol L^−1^ of AuCl_4_^−^ are the most suitable for the enzymatic polymerization of PANI- and Ppy-based nanocomposites. The highest rate of PANI/AuNPs-GOx and Ppy/AuNPs-GOx formation was observed in the solution of SA buffer, pH 6.0. The most optimal duration of the enzymatic synthesis of nanocomposites was within 4.5 days. In present our research enzymatically formed nanocomposites are attractive, because any other chemical compounds except glucose, GOx, AuNPs or AuCl_4_^−^ and corresponding monomer (aniline or pyrrole), which forms conducting polymer matrix, are used in the polymerization course.

## Figures and Tables

**Figure 1 polymers-11-00377-f001:**
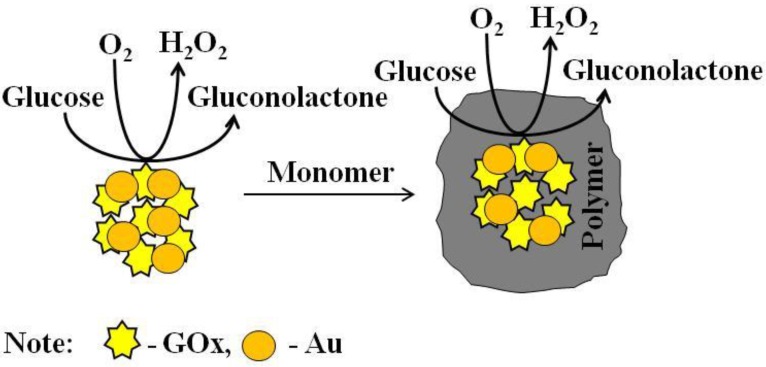
The formation of polymer/AuNPs-GOx-based nanocomposites during enzymatic polymerization.

**Figure 2 polymers-11-00377-f002:**
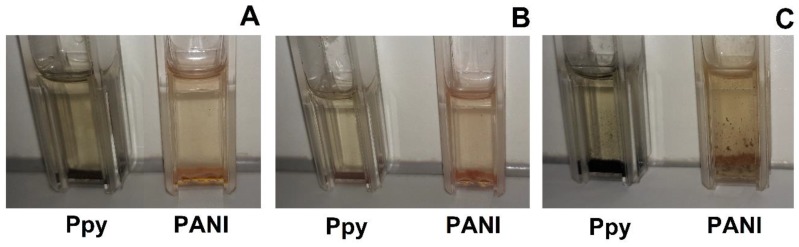
Samples of PANI/AuNPs-GOx and Ppy/AuNPs-GOx nanocomposites formed in the presence of AuNPs_(6nm)_ (**A**), AuNPs_(13nm)_ (**B**) and AuCl_4_^−^ (**C**) after 3 days lasting polymerization. Polymerization bulk solution was based on 0.05 mol L^−1^ SA buffer, pH 6.0, with 0.05 mol L^−1^ of glucose, 0.50 mol L^−1^ of aniline or pyrrole, 0.75 mg mL^−1^ of GOx and AuNPs_(6nm)_ (**A**), AuNPs_(13nm)_ (**B**) and AuCl_4_^−^ (**C**).

**Figure 3 polymers-11-00377-f003:**
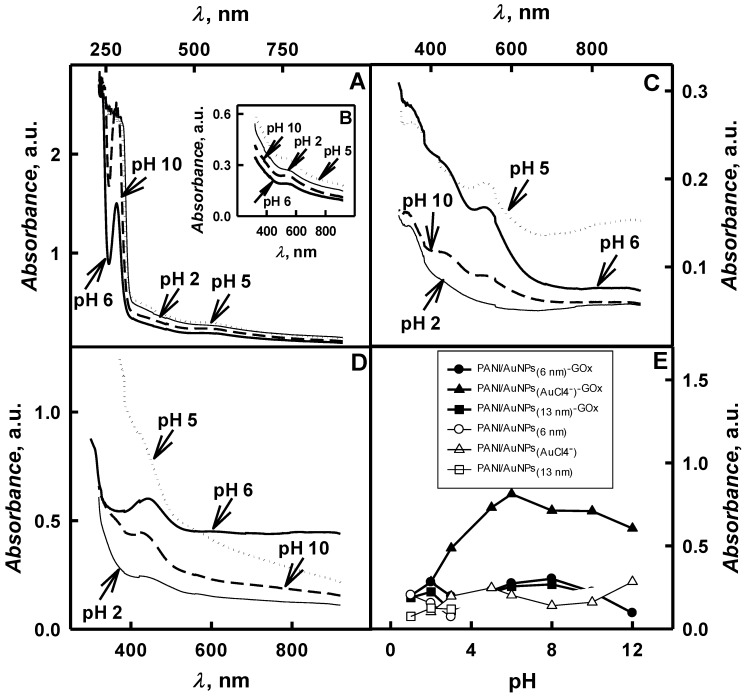
Spectra of PANI/AuNPs-GOx formed in the presence of AuNPs_(6nm)_, AuNPs_(13nm)_ and AuCl_4_^−^ (**A**,**B**,**C**,**D**) and the influence of pH on the absorbance maximum at *λ* = 450 nm during the enzymatic polymerization and auto-polymerization in the absence of GOx (**E**). (Polymerization solution composition: 0.05 mol L^−1^ glucose, 0.50 mol L^−1^ aniline, 0.75 mg mL^−1^ glucose oxidase and AuNPs_(6nm)_ (**A**,**B**), AuNPs_(13nm)_ (**C**) and 0.6 mmol L^−1^ AuCl_4_^−^ (**D**); 2 days of enzymatic polymerization. The wavelength interval of spectra in **B** is in the range from 320 to 900 nm. Optical absorbance (**E**) was registered in 0.05 mol L^−1^ SA buffer, pH 6.0).

**Figure 4 polymers-11-00377-f004:**
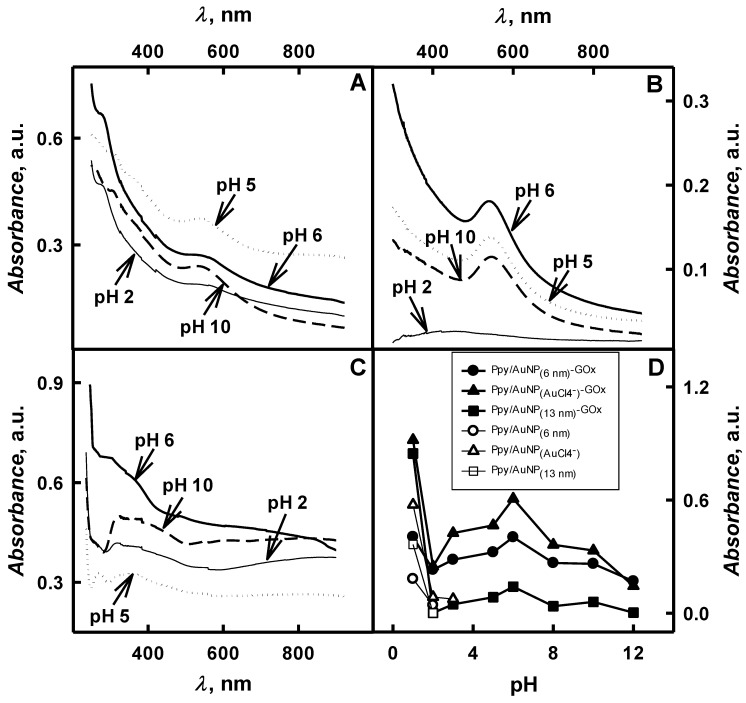
Spectra of Ppy/AuNPs-GOx formed in the presence of various forms of gold sources (**A**,**B**,**C**) and the influence of pH on the absorbance maximum at *λ* = 480 nm during enzymatic polymerization and auto-polymerization in the absence of enzyme (**D**). (Polymerization solution composition: 0.05 mol L^−1^ glucose, 0.50 mol L^−1^ pyrrole, 0.75 mg mL^−1^ glucose oxidase and AuNPs_(6nm)_ (**A**), AuNPs_(13nm)_ (**B**) and 0.6 mmol L^−1^ AuCl_4_^−^ (**C**); 2 days of enzymatic polymerization. Optical absorbance (**D**) was registered in 0.05 mol L^−1^ SA buffer, pH 6.0).

**Figure 5 polymers-11-00377-f005:**
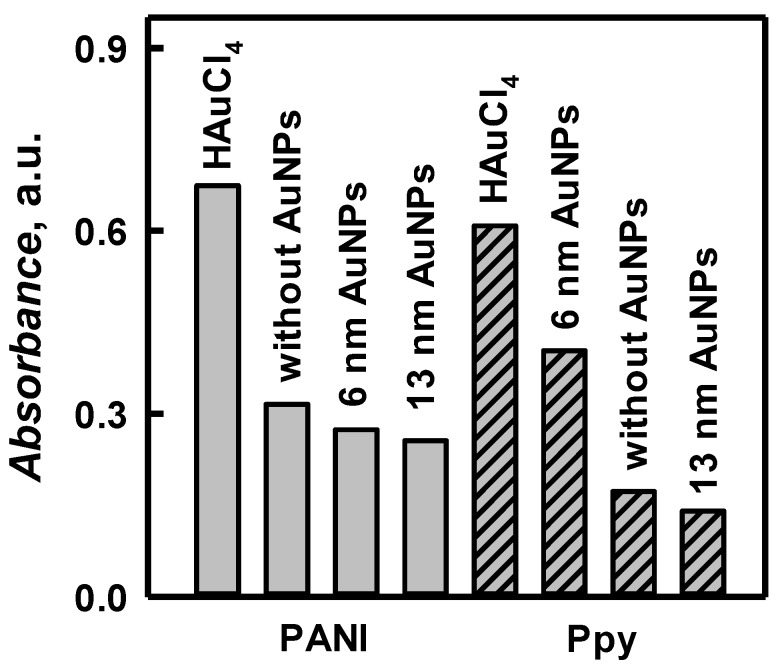
The absorbance diagram of PANI/GOx- and Ppy/GOx-based nanocomposites in the presence and absence of AuNPs. (Composition of polymerization bulk solution is presented in chapter 2.3.; optical absorbance was registered in 0.05 mol L^−1^ SA buffer, pH 6.0, at *λ* = 450 nm for PANI-based samples and at *λ* = 480 nm for Ppy-based samples.)

**Figure 6 polymers-11-00377-f006:**
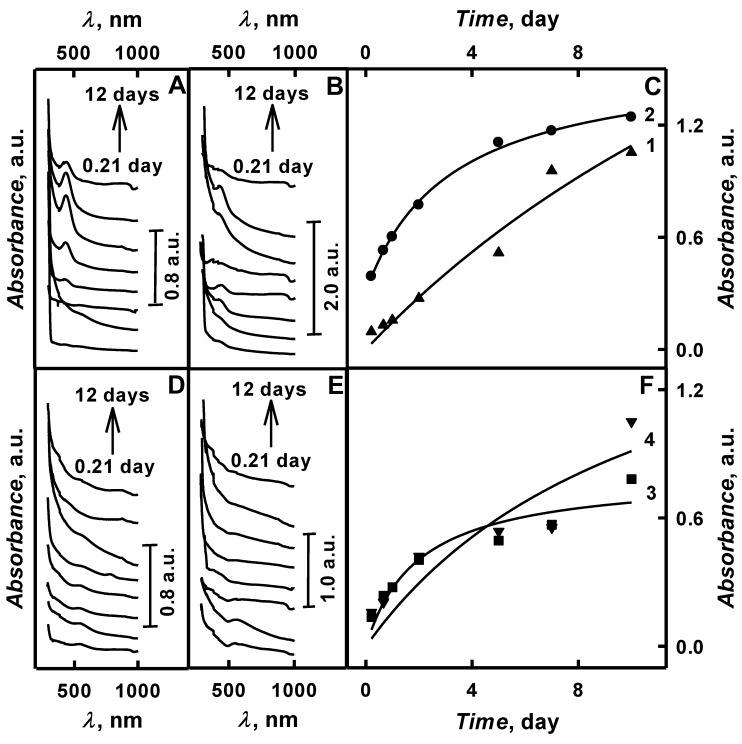
Spectra registered during enzymatic synthesis of aniline (**A**,**B**) and pyrrole (**D**,**E**) in the presence of various gold-containing compounds and the influence of polymerization duration on optical absorbance of formed nanocomposite colloidal solutions (**C**,**F**). (Polymerization solution composition: 0.05 mol L^−1^ glucose, 0.50 mol L^−1^ aniline (**A**,**B**,**C**) or pyrrole (**D**,**E**,**F**), 0.75 mg mL^−1^ glucose oxidase and AuNPs_(6nm)_ (**A**,**D**) or 0.6 mmol L^−1^ AuCl_4_^−^ (**B**,**E**). Optical absorbance was registered in 0.05 mol L^−1^ SA buffer, pH 6.0. (**C**) 1,2 curves – PANI/AuNPs_(6nm)_-GOx and PANI/AuNPs_(AuCl_4__^−^_)_-GOx; (**F**) 3,4 curves – Ppy/AuNPs_(6nm)_-GOx and Ppy/AuNPs_(AuCl_4__^−^_)_-GOx.).

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
