# Peer review of "Formation of Polyaniline and Polypyrrole Nanocomposites with Embedded Glucose Oxidase and Gold Nanoparticles"

_polymers, 2019, doi:10.3390/polym11020377_

Round 1

Reviewer 1 Report

Considering the prior scientific production of the Authors, there is not particular novelty in the paper under evaluation, but it is a meticulous study on enzymatic polymerization based formation of polyaniline and polypyrrole  nanocomposites with embedded glucose oxidase and gold nanoparticles.

In general, the experiments that have been carried out demonstrate good evidence and the results are clear.

fig 3b have to be better explain in the caption

fig 3E and 4D are not easy readable, I suggest to added legends and/or a better explaination in the captions

The paper is suitable for publication to this journal after minor revision.

Author Response

Response to reviewer #1:

We would like to thank the reviewer for very professional review of our manuscript, valuable comments and recommendations. Thank you for pointing out our mistakes and giving suggestions which further on improve clarity of this paper. We did our best in order to improve the manuscript according to comments and recommendations. All the most important changes are highlighted in the revised manuscript. Corrections and changes are highlighted in the manuscript (in red).

Please find below   short explanations and answers to your questions:

Reviewer #1 wrote:  Considering the prior scientific production of the Authors, there is not particular novelty in the paper under evaluation, but it is a meticulous study on enzymatic polymerization based formation of polyaniline and polypyrrole  nanocomposites with embedded glucose oxidase and gold nanoparticles. In general, the experiments that have been carried out demonstrate good evidence and the results are clear.

Response to Reviewer #1: We will thank the reviewer for highly insightful comments.

Reviewer #1 wrote: fig 3b have to be better explain in the caption

Response to Reviewer #1: Fig 3B Some additional explanations are added to the caption: „The wavelength interval of spectra in B is in the range from 320 to 900 nm.”

Reviewer #1 wrote: fig 3E and 4D are not easy readable, I suggest to added legends and/or a better explaination in the captions

Response to Reviewer #1: Fig 3E and 4D now are better explained by the addition of legends.

Reviewer #1 wrote: The paper is suitable for publication to this journal after minor revision.

Response to Reviewer #1: We will thank the reviewer for highly insightful comments and kind recommendation.

We will thank for positive feedback and valuable recommendations.

We hope after all these corrections our manuscript is suitable for publication.

Yours sincerely,
Arunas Ramanavicius

----------------------------------------------------------------
Prof. habil. dr. Arunas Ramanavicius

Head of Department of Physical Chemistry,

Faculty of Chemistry, Vilnius University,

Naugarduko 24, 03225 Vilnius 6, Lithuania; e-mail: [email protected]

Reviewer 2 Report

This manuscript describes the formation of gold nanoparticle (AuNP) and polymer composite through the in situ enzymatic polymerizations of polyaniline (PANI) and polypyrrole (Ppy). The influence of polymerization composition and pH on the formation of PANI- and Ppy-based nanocomposites was investigated. However, the lack of characterizations for nanocomposites weaken the manuscripts. Below are some specific comments.

1.    The author should include the control group for the enzymatic polymerization of PANI and Ppy in the absence of AuNP as well as the AuNP in glucose solution without aniline or pyrrole monomer.

2.    UV/Vis spectroscopy was used as the exclusive method to characterization the polymerization. The nanocomposites underwent a phase separation in the presence of AuNP and post the enzymatic polymerizations. The author should also provide the characterization for the nanocomposites obtained. Composition and morphology analysis of the produced nanocomposites should be conducted.

3.    The author attempted to investigate the effects of polymerization kinetics on the formation of the nanocomposites (Fig. 6 and corresponding discussion). The author concluded that the prolonged polymerization time leads to the formation of large nanocomposites (Page 11, line 326). However, there is no evidence to support the comments.

Author Response

Response to reviewer #2:

We would like to thank the reviewer for very professional review of our manuscript, valuable comments and recommendations. Thank you for pointing out our mistakes and giving suggestions which further on improve clarity of this paper. We did our best in order to improve the manuscript according to comments and recommendations. All the most important changes are highlighted in the revised manuscript. Corrections and changes are highlighted in the manuscript (in red).

Please find below   short explanations and answers to your questions:

Reviewer #2 wrote: This manuscript describes the formation of gold nanoparticle (AuNP) and polymer composite through the in situ enzymatic polymerizations of polyaniline (PANI) and polypyrrole (Ppy). The influence of polymerization composition and pH on the formation of PANI- and Ppy-based nanocomposites was investigated. However, the lack of characterizations for nanocomposites weaken the manuscripts. Below are some specific comments.

Response to Reviewer #2: We will thank the reviewer for highly insightful comments.

Reviewer #2 wrote: The author should include the control group for the enzymatic polymerization of PANI and Ppy in the absence of AuNP as well as the AuNP in glucose solution without aniline or pyrrole monomer.

Response to Reviewer #2: The control group for the enzymatic polymerization of PANI and Ppy in the absence of AuNPs was included in figure 5. The comparison of optical absorbance for enzymatically synthesized polymeric nanoparticles in the presence and absence of AuNPs was performed and recently it is described in manuscript.

Figure 5. The absorbance diagram of PANI/GOx- and Ppy/GOx-based nanocomposites in the presence and absence of AuNPs. Composition of polymerization bulk solution is presented in chapter 2.3.; optical absorbance was registered in 0.05 mol L-1 SA buffer, pH 6.0, at l=450 nm for PANI-based samples and at l=480 nm for Ppy-based samples.

During control experiments we have tested: (i) the polymerization bulk solutions of PANI and Ppy, which not contained AuNPs and (ii) the solution, which contained glucose, GOx, AuNPs but not contained any monomers (pyrrole or aniline), which are required for the formation of polymers. As it is presented in figure 5 the optical absorbance of PANI/AuNPs(AuCl4-)-GOx and Ppy/AuNPs(AuCl4-)-GOx nanocomposites was 2.14 and 3.52 times higher if compared with that registered for PANI/GOx (0.315 a.u) [9] and Ppy/GOx (0.173 a.u.). The formation rate for PANI/AuNPs(6nm)-GOx and PANI/GOx was very similar; for Ppy/AuNPs(6nm)-GOx was 2.33 times higher than that obtained for Ppy/GOx nanocomposites. The formation rate of PANI/AuNPs-GOx and Ppy/AuNPs-GOx nanocomposites in comparison to that of PANI/GOx and Ppy/GOx is significantly higher, because AuNPs speed-up the agglomeration of formed oligomers and the most probably facilitate polymerization reaction [36,37]. As it was expected any observable composites were formed in the solution containing glucose, GOx and AuNPs in the absence of PANI and Ppy. It means that in enzymatic polymerization at least three compounds (glucose, GOx and aniline or pyrrole) are required for the formation of composite materials. Therefore, for further investigations of PANI/AuNPs-GOx and Ppy/AuNPs-GOx nanocomposite formation AuNPs(6nm) and AuCl4- were chosen, which were both considered to be the most suitable for the formation of these pyrrole- and aniline-based nanocomposites [30].

Reviewer #2 wrote: UV/Vis spectroscopy was used as the exclusive method to characterization the polymerization. The nanocomposites underwent a phase separation in the presence of AuNP and post the enzymatic polymerizations. The author should also provide the characterization for the nanocomposites obtained. Composition and morphology analysis of the produced nanocomposites should be conducted.

Response to Reviewer #2: We are thankful for this recommendation, recently additional research on the characterization and possible bioanalytical application of in present manuscript reported PANI/AuNPs-GOx and Ppy/AuNPs-GOx nanocomposites is performed and these findings will be reported in next coming manuscript. Below we are presenting the description of preliminary results of some the most recently performed characterization experiments, which will be finished soon and published within short time:

‘Some scientists evaluated in their papers the size of different synthesized polymer/gold nanocomposites by SEM. Venditti et. al formed and investigated PANI/AuNPs nanostrtuctured composites with 180-220 nm of size [27]. Berzina and et. al. described the oxidative formation of polyaniline/gold with nanoparticles of 20 nm of size [37]. During the enzymatic polymerization of aniline using chitosan and poly(N-isopropylacrylamide) as steric stabilizers were synthesized 50 nm of PANI nanoparticles [30]. Polyaniline matrix with small anions is characterized by poor stability. To solve this problem large anions are added in aniline polymerization solution [6]. The size of particles in solution is mostly determined by hydrodynamic diameter. The hydrodynamic diameter of enzymatically synthesized polyaniline and polypyrrole nanocomposites with embedded glucose oxidase and gold nanoparticles was evaluated by dynamic light scattering (DLS) technique. The formation of PANI/AuNPs-GOx and Ppy/AuNPs-GOx was performed in the solution of 0.05 mol L-1 SA buffer, pH 6.0, with 0.50 mol L-1 aniline or pyrrole, 0.75 mg mL-1 GOx and 0.05 mmol L-1 glucose in the presence of 6 nm (26.0 nmol L-1) AuNPs or 0.6 mmol L-1 HAuCl4 during 4.5 days at room temperature (+20 ± 2°C). After enzymatic synthesis of PANI/AuNPs-GOx and Ppy/AuNPs-GOx particles were separated, washed and collected by a centrifugation as was described in chapter 2.2 of manuscript. Hydrodynamic diameter of formed after 4.5 days PANI/AuNPs-GOx and Ppy/AuNPs-GOx structures in the presence of 6 nm AuNPs or 0.6 mmol L-1 HAuCl4 was evaluated by Zetasizer Nano ZS from Malvern (Herrenberg, Germany) device equipped with a 633 nm He-Ne laser and operating at 173° angle using dynamic light scattering technique. The obtained DLS data was analysed with Dispersion Technology Software version 6.01 from Malvern (United Kingdom).

The distribution of Ppy nanocomposites with embedded GOx and AuNPs hydrodynamic diameter after 4.5 days-polymerization is presented below. Hydrodynamic diameter of PANI/AuNPs(6 nm)-GOx, PANI/AuNPs(AuCl4-)-GOx, Ppy/AuNPs(6 nm)-GOx and Ppy/AuNPs(AuCl4-)-GOx composites was evaluated as 1128, 659, 594 and 388 nm, respectively. It is seen that formed PANI/AuNPs(6 nm)-GOx and Ppy/AuNPs(6 nm)-GOx particles were 1.71 and 1.70 times higher if compared with PANI/AuNPs(AuCl4-)-GOx and Ppy/AuNPs(AuCl4-)-GOx. It was noticed that hydrodynamic diameter of formed after 4.5 days PANI/AuNPs(6 nm)-GOx and PANI/AuNPs(AuCl4-)-GOx particles was 9.81 and 5.73 times higher than that characterized for previously investigated PANI-GOx (115 nm) after the same polymerization time [9].

Figure. The distribution of Ppy nanocomposites with embedded GOx and AuNPs hydrodynamic diameter after 4.5 days-polymerization. The composition of polymerization solution: 0.05 mol L-1 SA buffer, pH 6.0, with 0.05 mol L-1 glucose, 0.75 mg mL-1 glucose oxidase and 0.50 mol L-1 pyrrole, respectively. DLS signal was registered in 0.05 mol L-1 SA buffer, pH 6.0.

Reviewer #2 wrote: The author attempted to investigate the effects of polymerization kinetics on the formation of the nanocomposites (Fig. 6 and corresponding discussion). The author concluded that the prolonged polymerization time leads to the formation of large nanocomposites (Page 11, line 326). However, there is no evidence to support the comments.

Response to Reviewer #2: The formation of the polymer composites (oligomers) is time-limited process, hence long polymerization time contributes to the increase of polymeric particle size [9]. It could be explained by the formation of thicker and less transparent films when the polymerization is performed longer. During enzymatic polymerization monomers are being oxidized and increasing the length of oligomeric and/or polymeric chains. Such oligomers and polymers aggregates around the exciting polymers and adsorbs on the AuNPs. According to experiments, which will be soon published elsewhere: ‘’during long-lasting polymerization some agglomerates characterized by high hydrodynamic diameter were formed. The formation of PANI/AuNPs-GOx and Ppy/AuNPs-GOx was performed in the solution of 0.05 mol L-1 SA buffer, pH 6.0, with 0.50 mol L-1 aniline or pyrrole, 0.75 mg mL-1 GOx and 0.05 mmol L-1 glucose in the presence of 6 nm AuNPs or 0.6 mmol L-1 HAuCl4 from 1 to 7 days at room temperature. Hydrodynamic diameter of enzymatically synthesized PANI/AuNPs-GOx and Ppy/AuNPs-GOx oligomers were evaluated by dynamic light scattering technique and it was depended on polymerization time. Hydrodynamic diameter of PANI/AuNPs(6 nm)-GOx, PANI/AuNPs(AuCl4-)-GOx, Ppy/AuNPs(6 nm)-GOx and Ppy/AuNPs(AuCl4-)-GOx particles was evaluated as 679, 570, 511 and 366 nm after 1 days of polymerization. By an increase of polymerization duration until 4.5 days in the presence of 6 nm AuNPs or 0.6 mmol L-1 HAuCl4 hydrodynamic diameter of PANI/AuNPs(6 nm)/GOx and PANI/AuNPs(AuCl4-)/GOx particles was increased until 1128 and 659 nm; hydrodynamic diameter of Ppy/AuNPs(6 nm)-GOx and Ppy/AuNPs(AuCl4-)-GOx particles – until 594 and 388 nm. It is seen that the hydrodynamic diameter of formed PANI/AuNPs(6 nm)-GOx, PANI/AuNPs(AuCl4-)-GOx, Ppy/AuNPs(6 nm)-GOx and Ppy/AuNPs(AuCl4-)-GOx particles after 4.5 days of the polymerization was 1.66, 1.16, 1.16 and 1.06 times higher if compared with results obtained after 1 day. In the comparisons with previously presented research measurements, where the size was varied between 680 and 1080 nm after one and three days of polymerization, the size of our created pyrrole/gold composites after 4.5 days of polymerization was higher [36]. Therefore, for future investigations not longer than 4.5 days lasting enzymatic polymerization has been performed.’’

We will thank for positive feedback and valuable recommendations.

We hope after all these corrections our manuscript is suitable for publication.

Yours sincerely,
Arunas Ramanavicius

----------------------------------------------------------------
Prof. habil. dr. Arunas Ramanavicius

Head of Department of Physical Chemistry,

Faculty of Chemistry, Vilnius University,

Naugarduko 24, 03225 Vilnius 6, Lithuania; e-mail: [email protected]

Round 2

Reviewer 2 Report

No further revision is needed.